# Evaluation of the Reactivity of Methanol and Hydrogen Sulfide Residues with the Ziegler–Natta Catalyst during Polypropylene Synthesis and Its Effects on Polymer Properties

**DOI:** 10.3390/polym15204061

**Published:** 2023-10-12

**Authors:** Joaquín Hernández-Fernández, Rafael González-Cuello, Rodrigo Ortega-Toro

**Affiliations:** 1Chemistry Program, Department of Natural and Exact Sciences, San Pablo Campus, University of Cartagena, Cartagena 130015, Colombia; 2Chemical Engineering Program, School of Engineering, Universidad Tecnológica de Bolivar, Parque Industrial y Tecnológico Carlos Vélez Pombo, Km 1 Vía Turbaco, Turbaco 130001, Colombia; 3Department of Natural and Exact Science, Universidad de la Costa, Barranquilla 30300, Colombia; 4Food Packaging and Shelf-Life Research Group (FP & SL), Food Engineering Program, Universidad de Cartagena, Avenida del Consulado St. 30, Cartagena de Indias 130015, Colombia; rgonzalezc1@unicartagena.edu.co (R.G.-C.); rortegap1@unicartagena.edu.co (R.O.-T.)

**Keywords:** Ziegler–Natta polymerization, hydrogen sulfide, methanol, catalyst productivity, molecular weight, melt flow index, xylene solubility, DFT calculations, reactivity descriptors

## Abstract

The study focused on the evaluation of the influence of inhibitory compounds such as hydrogen sulfide (H_2_S) and methanol (CH_3_OH) on the catalytic productivity and properties of the polymers in the polymerization process with the Ziegler–Natta catalyst. The investigation involved experimental measurements, computational calculations using DFT, and analysis of various parameters, such as molecular weight, melt flow index, xylene solubility, and reactivity descriptors. The results revealed a clear correlation between the concentration of H_2_S and methanol and the parameters evaluated. Increasing the H_2_S concentrations, on average by 0.5 and 1.0 ppm, resulted in a drastic decrease in the polymer’s molecular weight. A directly proportional relationship was observed between the flow rate and the H_2_S concentration. In the case of methanol, the change occurred from 60 ppm, causing a sharp decrease in the molecular weight of the polymer, which translates into an increase in the fluidity index and a decrease in solubility in xylene. The presence of these inhibitors also affected the catalytic activity, causing a reduction in the productivity of the Ziegler–Natta catalyst. Computational calculations provided a deeper understanding of the molecular behavior and reactivity of the studied compounds. The computational calculations yielded significantly lower results compared to other studies, with values of −69.0 and −43.9 kcal/mol for the Ti-CH_3_OH and H_2_S interactions, respectively. These results indicate remarkable stability in the studied interactions and suggest that both adsorptions are highly favorable.

## 1. Introduction

In 1953, scientists Karl Ziegler and Giulio Natta introduced the Ziegler–Natta (Z-N) catalyst, based on organometallic compounds, after discovering its ability to catalyze the polymerization of olefins such as ethylene and propylene. Z-N is used in the coordinated polymerization process, where an alkylaluminum compound in the catalyst plays the initiator role. In contrast, a titanium compound (TiCl_2_) acts as a propagation center, allowing the formation of high-quality polymers and controlling the structure and properties of these macromolecules [1,2,3]. The Z-N catalyst is composed of several key elements, such as MgCl_2_, TiCl_4_, trialkylaluminum, and an electron donor. MgCl_2_ is commonly used as catalyst support, providing a solid structure for the active components [1]. TiCl_4_ acts as a catalyst precursor and can react with trialkylaluminum to form an active TiCl_2_R catalyst, where R is an alkyl group [4,5]. The presence of the electron donor is essential since it acts as a Lewis base, increasing the stereoselectivity of the catalyst and affecting the properties and characteristics of the resulting polymer. However, it must be taken into account that the Z-N catalyst may modify the distribution of the active sites, which may have various consequences, including the sensitivity of the catalyst to hydrogen. Considering these factors when using the Z-N catalyst in polymerization is essential to achieve optimal control of polymer properties [6,7,8].

One of the factors by which the propylene polymerization process is affected in the presence of the Z-N catalyst is the existence of polluting compounds such as ketones, alcohols, and gases, among others [9,10]. The presence of impurities in trace amounts can significantly impact the polymerization process. These impurities, such as alcohols, act as inhibitors and can interfere with critical stages of the process [11,12,13,14]. Alcohols, by functioning as Lewis bases, can interact with the catalyst and decrease its ability to activate and propagate polymer molecules. As a result, the catalytic activity is affected, which can cause changes in the rate and efficiency of the reaction [15]. Studies have been carried out to better understand the behavior of alcohols as inhibitors in the polymerization of propylene and their effect on the performance and properties of the resulting polymer. One such study was conducted by Tangjituabun et al. using methods of suspension polymerization, interrupted flow polymerization, and GPC analysis. During the polypropylene production process, alcohols and ketones were used as inhibitors. The characterization of the obtained polymers was carried out, and the results revealed that a molar ratio of inhibitory materials/Ti of 0.1 was found in the suspension polymerization. A significant decrease in catalyst activity was observed, with acetate having the main responsibility for this reduction, followed to a lesser extent by the presence of acetone and an even lesser extent by the influence of methanol [16,17]. In this context, it is relevant to highlight that methanol, acetone, and acetate are not the only compounds in the polymerization process that can affect the final result [18,19,20,21,22,23]. The inorganic compound hydrogen sulfide (H_2_S) has a differentiated impact compared to methanol. Its weakly acidic nature characterizes H_2_S, and it exhibits the ability to form insoluble metal sulfides by binding to transition metals (M). This compound is of particular interest in the catalysis of polymeric materials due to its organometallic properties—among which the formation of vacant coordinate sites stands out—which are essential for understanding the reactivity and behavior in catalysis. Its effect is highlighted in the alteration of the properties of the resulting polymer. The presence of H_2_S in propylene can be attributed to failures during its production process, and in some cases, it is considered a polymerization inhibitor [11].

This research focuses on studying two specific compounds, methanol and H_2_S, as inhibitors of the Ziegler–Natta catalyst in polypropylene production. There are very few studies on methanol as an inhibitor, and including H_2_S as an inhibitor is a novel aspect in this context. H_2_S has organometallic properties that can have a differential impact on the catalysis of polymeric materials and can significantly alter the properties of the resulting polymer. Furthermore, the research is not only limited to identifying the presence of inhibitors but also carried out extensive tests and analyses to assess their influence on catalytic activity during polymerization. Important properties such as melt flow rate, catalyst productivity, and molecular weight distribution of polypropylene are discussed to reveal how inhibitors affect the polymerization process and final product characteristics. Computational calculations using the density functional theory (DFT) method and UKA Fukui were also incorporated to determine reactivity profiles. This provides additional information on the mechanisms and reactions that occur in the presence of the inhibitors. Understanding how inhibitors affect the polymerization process and the properties of the resulting polymer allows for the optimization of polymer production. By identifying and minimizing the presence of inhibitors, the efficiency and quality of the polymers produced can be improved, directly benefiting the chemical industry. By knowing the mechanisms and reactions that occur during polymerization in the presence of inhibitors, more efficient and controlled techniques for polymer production can be developed. This can lead to more profitable and sustainable manufacturing processes. Research in producing high-quality polymers and reducing impurities and contaminants such as inhibitors has a positive impact on sustainable development. By improving polymer manufacturing processes, waste, resource consumption, and the environmental impact of chemical production can be reduced. Research in this field not only benefits the industry but also contributes to the advancement of scientific knowledge in the field of catalysis and polymerization. Understanding how inhibitors affect polymer properties at the molecular level provides valuable information for future research and the development of new catalysts and processes. The results of this research are not limited to the production of polypropylene; the knowledge gained on catalysis and polymerization can also be applied to other fields of chemistry and the materials industry, broadening the scope of the impact of the research.

## 2. Materials and Methods

### 2.1. Reagents and Materials

H_2_S, supplied by Merk in Darmstadt, Germany and with a purity of 99.99%, and 99.9% methanol, provided by Merck KGaA, Darmstadt, Germany, were used. Polypropylene (PP), supplied by Ecopetrol Petrochemical in Cartagena, Colombia, was also used. The catalyst used was a fourth-generation Ziegler–Natta with a MgCl_2_ support. In addition, triethylaluminum (TEAL) provided by Merk in Germany and with a purity of 98%, was used diluted in n-heptane. n-Dibutyl phthalate (hereinafter DBP) was used as an internal electron donor and was provided by Sigma-Aldrich Inc., Saint Louis, MO, USA, EE.UU.

### 2.2. Polymerization

The gas-phase polymerization process followed the PP synthesis method using Ziegler–Natta catalysts, as described by Hernández [19]. The procedure was carried out in a fluidized bed reactor previously purged with nitrogen for conditioning. Next, 1.3 MT h^−1^ and 31 g h^−1^ of hydrogen were introduced from the bottom of the reactor. Likewise, 6 kg h^−1^ of ZN catalyst, 0.26 kg h^−1^ of co-catalyst, and 1.1 mol h^−1^ of selectivity control agent were added together with a stream of nitrogen. This process was carried out in discontinuous mode (batch) at a temperature of 70 °C and a pressure of 27 bar, as shown in Figure 1. The addition of H_2_S and methanol was carried out in the propylene supply line. Subsequently, the hydrocarbon residues were removed from the resin using a stream of nitrogen and steam, thus obtaining a high-quality virgin resin. It should be noted that each process was carried out individually for methanol and H_2_S.

Samples of the final polypropylene obtained were taken, as described in detail in Table 1.

### 2.3. Melt Flow Index—MFI

A Tinius Olsen MP1200 plastometer was used to measure the melt flow index (*MFI*). The plastometer maintained a constant temperature of 240 °C in its cylinder, and a 2.20 kg piston was used to displace the melt. After obtaining the *MFI* data, the average molecular weight of each polypropylene sample was evaluated using a method known as the Bremner approximation.
(1)$$M_{W}=\frac{1675}{\left(\left(MFI\;230\;{}^{\circ}C\;2.16\;kg\right)\times 10^{-21}\right)}\;$$

### 2.4. Determination of the Soluble Fraction in Xylene

The results of this test provide a way to measure the relative amount of polypropylene polymers that are soluble in xylene. The xylene soluble fraction is roughly related to the amorphous portion of polypropylene. The use of xylene is preferred to determine this soluble fraction since it is more specific for detecting the atactic amount of polymers than other solvents. It has been observed that the concentration of the soluble fraction obtained through the use of the particular solvent (xylene) correlates closely with the product’s performance characteristics in specific applications, such as in the manufacture of films or fibers. In other words, the amount of soluble fraction can significantly impact how the polymer performs in different end uses.

In this investigation, the amount of fraction soluble in xylene (XS) was measured at a temperature of 25 °C using the procedure established in ISO 16152 [24].

### 2.5. Computational Details

All calculation, optimization, and frequency analysis operations were performed using the density functional theory (DFT) via Gaussian 16 software, Revision C.01. The B3LYP functional method with the 6–311G(d,p) basis set was employed to refine the structures of inhibitors and the catalyst. Dispersion corrections were considered using the DFT-D3 method (without damping), and harmonic vibrational frequency calculations were conducted to verify that the structures were optimized correctly, following the methodology proposed by Xing Guo [1]. In the β-MgCl_2_(110) surface model, all atoms were held fixed except for the two chlorine atoms directly bonded to the titanium center. All other atoms were adjusted during the calculations to allow for relaxation.

The adsorption energy of the inhibitors was calculated according to Equation (2).
(2)$$E_{ad}=\frac{E_{cat/Inhibitor}-E_{cat}-nE_{Inhibitor}}{n}\;\;\;$$
where $E_{cat/Inhibitor}$ is the adsorption energy of the inhibitor absorbed at the active site, and $E_{cat}$ and $E_{Inhibitor}\;$ are the energies of the isolated active center and the isolated inhibitor molecule, respectively; *n* is the number of inhibitory molecules.

The optimized structures were visualized using Gaussview06 software, Revision C.01 (Gaussian, Inc., Wallingford, CT, USA) and analysis was conducted on the highest occupied molecular orbital (HOMO) and lowest unoccupied molecular orbital (LUMO). The energy values of these orbitals were used to explore global-level chemical reactivity properties such as electronegativity (χ) and chemical potential (μ). Additionally, characteristics like global softness (S), nucleophilicity index (N) (using tetracyanoethylene (TCE) as a reference), electrophilicity index (ω), and hardness (η) were examined.

A molecular electrostatic potential (MEP) map was also generated to visualize the electrical charge distribution of each of the compounds of interest. In addition, the UKA FOKUI 2.00 software was used to calculate values of chemical descriptors and local reactivity properties. These calculations provide detailed information about the local characteristics and reactivity of methanol and H_2_S, providing a deeper understanding of their chemical behavior and properties. Simulations of the UV–vis spectra of the catalyst were performed; for this, simplified structural models of nanoclusters were proposed by Takasao et al. These patterns were obtained by selectively removing TiClx units from the proposed nanoclusters. Then, the simplified nanoclusters were reoptimized using density function theory (DFT) with the B3LYP level of view and the def2-TZVP base set. The following was considered regarding the proposed structural model: a Ti cation coordinated with six ligands on the MgCl_2_(110) surface, considered the most representative representation of Ti sites in ZN catalysts. This model was used as the basis for the simulation of the UV–vis spectrum. In Figure 2, the Mg atoms are represented in yellow, the Cl atoms in green, and the Ti atoms in grey. These models were obtained from the structures of TiCl_4_-coated MgCl_2_ nanoplates, previously determined by Takasao et al. Dashed lines indicate MgCl_2_ surfaces that are involved in TiClx chemisorption.

## 3. Results

### 3.1. Influence of Methanol and H_2_S on the Properties of PP

#### 3.1.1. Effects on the Melt Flow Index (MFI) and MW of the PP

Figure 3 present the results of evaluating the properties of the obtained polymer, which reveals an apparent, direct influence of the presence of H_2_S (hydrogen sulfide) and methanol on said properties. The data obtained indicate that as the concentration of the inhibitors (H_2_S and methanol) increases, significant changes are produced in the melt index and the average molecular weight of the polymer. Specifically, it was observed that an increase in H_2_S concentrations, on average by 0.5 and 1.0 ppm, resulted in a drastic decrease in the molecular weight of the polymer. This implies a shortening in the length of the polymer chains and an increase in the fluidity of the polymer, respectively. On the other hand, it was noted that, in the case of methanol, the change occurs from 60 parts per million, which also causes an abrupt decrease in the molecular weight of the polymer. These findings show a directly proportional relationship between the fluidity index and hydrogen sulfide concentration and methanol, respectively. Furthermore, the concentration of the inhibitors is inversely related to the average molecular weight (Mw), indicating a significant reduction in the resulting polymer chains. Moreover, these results support the negative influence of H_2_S and methanol on polymer properties, highlighting the importance of carefully considering and controlling the presence of these compounds during the polymerization process to ensure the desired properties of the final polymer. They also underscore the need for additional studies to better understand the underlying mechanisms and long-term effects of H_2_S and methanol exposure on polymers. These other studies may lead to improved formulation and development of polymers with optimized mechanical properties and excellent resistance to environmental conditions. In short, it is crucial to advance in understanding these mechanisms to promote the development of more efficient and sustainable polymers in the chemical industry.

#### 3.1.2. Effects on Lost Melt Flow Index (MFI) and Lost Productivity of Ziegler–Natta Catalyst

Figure 4 presents a detailed analysis of the relationship between the H_2_S and CH_3_OH concentration as inhibitors and the fundamental parameters in polymerization, such as MFI and Mw. These results provide critical information on the impact of these inhibitors on the properties of the resulting polymer.

A directly proportional correlation was observed between the H_2_S concentration and both evaluated parameters, indicating that H_2_S has a significant effect on the physical properties of the polymer. As the H_2_S concentration increases, further loss of catalyst productivity occurs, affecting the melt flow rate of the polymer. This implies that the presence of H_2_S causes alterations in the flowability of the polymer and can affect the quality of the final product.

In addition, it was observed that as methanol concentrations increase, the melt flow rate lost in the polymer also increases. This implies that methanol affects the extension of polymer chains, which affects the productivity of the process. These changes in melt flow rate and catalyst productivity demonstrate the need to exercise rigorous control over the presence of H_2_S and methanol during polymerization to ensure the quality and desired characteristics of the final polymer.

#### 3.1.3. Effects of Inhibitors on the Solubility in Xylene of PP

The analysis illustrated in Figure 5 examines the impact of methanol concentration as an inhibitor and H_2_S on the percentage solubility in xylene. The information on the effects of the concentration of inhibitors such as methanol and hydrogen sulfide on the percent solubility in xylene and the isotacticity of the resulting polymer has important implications for the type of polypropylene (PP) formed.

Percent xylene solubility measures the polymer’s ability to dissolve in xylene, a commonly used solvent in the characterization of isotactic polypropylenes. A high percentage of solubility in xylene indicates a greater capacity of the polymer to dissolve, which is related to a greater isotacticity. The more isotactic the polypropylene, the more ordered its structure and the better its mechanical properties. An inversely proportional relationship was observed between the H_2_S concentration and the percentage of solubility in xylene, which indicates that as the H_2_S concentration increases, there is a decrease in the solubility of the polymer in xylene. This same relationship is observed in the case of methanol since both inhibitors presented similar behaviors. These findings suggest that both H_2_S and methanol can affect the polymer’s structure and its ability to dissolve in xylene, decreasing said percentage, which implies a reduction in the isotacticity of the polymer. This suggests that polymerization in the presence of these inhibitors may result in the formation of polypropylene with lower isotacticity, which may have implications for the final properties of the polymer, such as its mechanical strength and stiffness.

It is important to note that the different types of polypropylene (isotactic, atactic, or syndiotactic) have other properties and applications. Isotactic polypropylene is the most desirable in many applications due to its high stiffness, strength, and thermal stability. Therefore, the decrease in the percentage of solubility in xylene and the possible reduction in the isotacticity of the polypropylene formed in the presence of inhibitors can hurt its properties and limit its usefulness in specific applications. In addition, the constant presence of a selective control agent in this study helps to modulate this effect, maintaining a high percentage of solubility in xylene at low concentrations of H_2_S and methanol, respectively. This implies that selective control agents can help keep properties and low solubility rates in xylene, which is relevant to maintaining a higher isotacticity in the resulting polymer [25,26].

These findings highlight the importance of carefully controlling the concentration of methanol and hydrogen sulfide inhibitors during polymerization. In addition, the importance of considering selective control agents to maintain properties and low percentages of solubility in xylene is emphasized, which contributes to maintaining greater isotacticity in the resulting polymer. These measures can improve the formulation and development of polymers with optimized isotacticity characteristics, which is essential to determine the type of polypropylene produced and its applicability in different industries.

#### 3.1.4. Effects of Inhibitors on Catalyst Productivity and Melt Flow Rate

Below are the graphs that show the results of the productivity of the Ziegler–Natta catalyst expressed in metric tons per kilogram (MT/Kg) about the melt flow index (MFI). These variables were plotted as a function of the concentration of two inhibitors, methanol (Figure 6a) and hydrogen sulfide (Figure 6b). By examining these graphs, an inverse relationship between methanol concentration and catalyst productivity can be seen. As the concentration of the inhibitor increases, productivity decreases, which is reflected in an increase in the melt index. This suggests that methanol affects the catalytic activity of the catalyst, resulting in a decrease in polymer production.

Similarly, an inverse relationship between hydrogen sulfide concentration and catalyst productivity is evident. As the H_2_S concentration increases, productivity decreases, and at the same time, an increase in the flow rate is observed. These effects indicate that hydrogen sulfide also affects the catalytic activity of the catalyst and results in decreased productivity.

It is important to note that these influences are more noticeable in the case of hydrogen sulfide (H_2_S) than in the case of methanol. For example, at deficient concentrations, such as 2.0 ppm H_2_S, a melt flow rate of 22.5 g/10 min and a 10 MT/Kg decrease in catalyst productivity are observed. This is similar to the reduction of catalyst productivity when a methanol concentration of 100 ppm is reached.

### 3.2. Frontier Molecular Orbitals and Global Structure–Activity Descriptors

Density functional theory (DFT) has successfully provided a theoretical foundation for explaining important chemical concepts such as electronegativity, hardness, and softness as well as more specific concepts like the Fukui function and local softness. By using the Koopman’s theorem, we can calculate the ionization potential and electron affinity of certain chemical substances, enabling us to obtain values for electronegativity and hardness.

When the energy of the highest occupied molecular orbital (HOMO) is higher, it signifies that the molecule is more prone to react with substances known as electrophiles. Conversely, a lower energy level in the lowest unoccupied molecular orbital (LUMO) is crucial for the molecule to react with substances known as nucleophiles.

This index measures how willing a chemical substance is to accept electrons. A highly reactive nucleophile will have a low value of ω, while a highly reactive electrophile will have a high value of ω. This new reactivity index helps us quantify how much energy is stabilized when a system acquires additional electrons from its surroundings.

According to Pearson’s theory, we can calculate the number of electrons transferred from the inhibiting molecule to the metal atom. In a reaction between two systems with different electronegativity levels (such as a metal surface and an inhibiting molecule), there is an electron flow from the less electronegative molecule to the more electronegative one until the chemical potentials are equalized. 

The inhibitory effect of a compound is generally attributed to the adsorption of the molecule on the metal surface. Adsorption can be physical (physisorption) or chemical (chemisorption) depending on the strength of adsorption. During chemisorption, one of the reactive species acts as an electron pair donor, and the other acts as an electron pair acceptor. The energy of the highest occupied molecular orbital (EHOMO) measures the tendency of a molecule to donate electrons [26]. High EHOMO values indicate the molecule’s tendency to donate electrons to suitable accepting molecules with vacant and low-energy molecular orbitals. Increasing EHOMO values facilitate adsorption, thereby enhancing inhibition efficiency by influencing the transport process through the adsorbed layer. Therefore, higher EHOMO values indicate a better electron-donating tendency, improving inhibitor adsorption on mild steel and consequently enhancing inhibition efficiency. ELUMO indicates the molecule’s capacity to accept electrons. The inhibitor’s binding ability to the metal surface increases with higher HOMO values and lower LUMO energy values. The frontier molecular orbital diagrams of H_2_S and methanol are shown in Figure 7. Table 2 represents the total energy and energy levels calculated in (eV) for the HOMO, LUMO, and energy gap of the investigated molecules.

According to the frontier molecular orbital (FMO) theory in chemical reactivity, electron transfer occurs due to the interaction between two types of orbitals: the highest occupied molecular orbital (HOMO) and the lowest unoccupied molecular orbital (LUMO) of reacting substances [27]. The EHOMO value is a chemical parameter often associated with a molecule’s electron-donating capacity. A high EHOMO value suggests that the molecule has a tendency to donate electrons to another molecule with a low-energy vacant orbital [28].

The inhibitor not only can donate electrons to the vacant (LUMO) orbital of the metal ion but can also accept electrons from the metal d orbital, creating a feedback type of bonding. According to our results, the highest EHOMO value, which is −7.1648 (eV) for methanol, indicates a more efficient capacity of the inhibitor to induce Ziegler–Natta catalyst productivity loss.

The energy difference, calculated as (ΔE = ELUMO − EHOMO), is a key factor influencing an inhibiting molecule’s ability to interact with the metal surface. When this energy difference decreases, it means the molecule is more reactive, resulting in higher efficiency as an inhibitor since less energy is required to participate in the necessary chemical reactions [29]. A molecule with a low-energy gap is more adaptable and is usually associated with higher chemical activity but lower long-term stability. The results in Table 2 show that H_2_S (7.8352 eV) has the lowest-energy gap, suggesting it could be more effective as an inhibitor, as supported by experimental results.

Ionization energy is a key indicator of the reactivity of chemical substances, whether they are atoms or molecules. When this energy is high, it signifies that substances are stable and do not readily react. Conversely, when it is low, it indicates high reactivity and a propensity to interact with other substances. In this case, the low ionization energy, which is 0.7929 (eV) for methanol, indicates that this compound is highly effective as an inhibitor (See Table 3).

Absolute hardness and softness are key properties for assessing the stability and reactivity of a molecule. Chemical hardness essentially tells us how resistant a molecule is to changes in its electron cloud when subjected to small alterations during a chemical reaction. A “hard” molecule has a large energy gap between its components, whereas a “soft” molecule has a small energy gap [30]. In our study, H_2_S exhibits low hardness with a value of 3.9176 (eV) compared to methanol, indicating a small energy gap, as observed in Table 2. For easier electron transfer, adsorption typically occurs in the part of the molecule where softness (S), which is a local property, has the highest value [31]. H_2_S has the highest softness value at 0.2552, albeit with a difference from methanol of only 0.0327.

According to Sanderson’s electronegativity equalization principle [32], H_2_S, with its high electronegativity (3.3221 eV), tends to quickly equalize its properties, suggesting it is less reactive and therefore has lower efficiency as an inhibitor. In Table 3, we see that H_2_S has a higher electronegativity than methanol. This implies that the electronegativity difference between the metal and the inhibitor is greater in the case of methanol compared to H_2_S.

Mulliken population analysis is used to identify the sites in a molecule where inhibitors attach and calculate how charge is distributed throughout the molecule [33]. The general consensus among experts is that the more negatively charged an atom in the molecule (known as a heteroatom) is, the more easily it can adhere to the metal surface through a donor–acceptor type of interaction [34]. The charge distribution diagram for the catalyst and methanol and the one for the catalyst and H_2_S is shown in Figure 8, where regions marked in green indicate positive charges within the structure, and negative charges are highlighted in red areas. Brighter shades indicate larger charge magnitudes. It is essential to consider the scenario where a molecule will receive a certain amount of charge at one location and release a certain amount of charge through the same or another location [35]. Parr and Yang proposed that a higher value of the Fukui function suggests greater reactivity [36]. Therefore, when the condensed Fukui function has a higher value, it means that a specific center in the molecule is more reactive.

### 3.3. Molecular Electrostatic Potential of H_2_S and Methanol

The molecular electrostatic potential (MEP) map is a valuable tool in investigating global molecular structure and reactivity, as it provides detailed information about the charge distribution and the availability of electrons in a molecule. This uses colors to represent the different regions of the molecule according to their electron density. In the MEP, the red areas indicate a higher electron density, suggesting the presence of nucleophilic sites in the molecule, that is, areas where the molecule has a high probability of donating electrons. On the other hand, the blue areas represent an electron deficiency, indicating the presence of electrophilic sites, where the molecule has a higher affinity for accepting electrons. These characteristics are fundamental to understanding the molecule’s chemical reactivity since they allow us to predict the interactions with other chemical species. By identifying the nucleophilic and electrophilic sites in the MEP, we can determine which areas of the molecule are most likely to participate in chemical reactions and how they can interact with other substances.

Figure 9 presents a three-dimensional representation of the electrostatic effect, which covers a range of values from −3.502 × 10^−2^ to 3.502 × 10^−2^.

According to Figure 9a, the most prone site for an electrophilic attack in the case of H_2_S is located around the sulfur, indicated by the red area. In contrast, the most susceptible site for a nucleophilic attack is the blue areas distributed around the hydrogen atoms. In Figure 9b, the MEP of methanol is shown in a range that varies from −6.288 × 10^−2^ to 6.288 × 10^−2^ eV, where the blue, green, and red colors indicate the regions with the most positive electrostatic potential, zero potential, and most negative electrostatic potential, respectively. The red and yellow regions are primarily located on the oxygen atom, which is the most reactive site for an electrophilic attack. On the other hand, the blue regions are found around the hydrogen atoms, which are the most reactive sites for a nucleophilic attack.

### 3.4. Local Chemical Reactivity Descriptors (Fukui Function)

The Fukui function is a valuable tool for identifying the most reactive sites in a molecule, either for nucleophilic or electrophilic reactions. Proposed by Parr and Yang in 1984, this function provides us with information about the changes in electron density at each site in the molecule.

By calculating the Fukui functions, we can determine which sites are most likely to accept or donate electrons. Sites with higher $f^{+}\left(r\right)\;$ values are considered favorable sites for nucleophilic attack, meaning they have a more remarkable ability to accept electrons. On the other hand, the sites with higher values of $f^{-}\left(r\right)\;$ are conducive to an electrophilic attack, indicating that they have a higher affinity to donate electrons. The dual descriptor f(r) is a better way to identify a reactive site in a molecule. It is defined as ∆f= $f^{+}-f^{-}$, providing an adequate distinction between nucleophilic and electrophilic attack in a precise region according to its sign. When ∆f is less than zero (∆f< 0), the site is favorable for electrophilic attack. On the other hand, when ∆f is more significant than zero (∆f> 0), the site is good for a nucleophilic attack. This approach allows us to more precisely identify the reactive sites in a molecule and predict what type of chemical reaction is most likely to occur. Calculations were performed, and tables containing the Fukui functions ($f^{0}$, $f^{+}$, and $f^{-}$) and the dual descriptor (∆f) for H_2_S were created, allowing us to better understand the chemical reactivity of this molecule.

The maximum values of the local descriptors of the Fukui function, such as $f^{-}$, $f^{+}$, and $f^{0},\;$ indicate the sites where electrophilic, nucleophilic, or free radical attack on H_2_S is most likely to occur. According to the calculations, the sites most susceptible to an electrophilic attack in hydrogen sulfide are the ones in which the sulfide atom shows greater reactivity. On the other hand, the sites most vulnerable to nucleophilic attack are hydrogen atoms. On the other hand, sulfur sites are most susceptible to free radical attack.

Table 4 shows the variation of ∆f depending on the atoms. The results reveal that, from the calculated values of ∆f, the sulfide atom (1) appears to be the site prone to an electrophilic attack since the value of ∆f= −0.2929. Meanwhile, hydrogen atoms (2–3) present positive ∆f values, indicating they are the most favorable sites for a nucleophilic attack, with ∆f  values of 0.1436 and ∆f of 0.1482, respectively.

Table 5 shows the specific values of the Fukui functions and the dual descriptor for each site in the methanol molecule. 

The calculations identified the sites most prone to electrophilic, nucleophilic, and free radical attacks in methanol. According to the results, the oxygen atom at position 5 is more susceptible to electrophilic and nucleophilic attack. Furthermore, it was found that the carbon at position one and the oxygen at position 5 are the sites most vulnerable to free radical attack (Figure 10). These findings are supported by Table 5, which shows the variations in the dual descriptor (∆f) about atoms. The results indicate that the oxygen atom at position 5 has a negative value of ∆f (−0.2800), indicating its higher propensity to electrophilic attack. On the other hand, the carbon at position 1 has a positive value of ∆f (0.2166), indicating that it is a more favorable site for a nucleophilic attack.

### 3.5. Study of the Adsorption of Methanol and H_2_S on the Active Ti Center of the Ziegler–Natta Catalyst

In this study, we investigated the role of methanol and H_2_S as inhibitors of the active titanium center in a scientifically relevant context. Our analysis focused on the adsorption energy (*E_ad_*) of these inhibitors and their ability to stabilize the active titanium center on the 110 surface of MgCl_2_ through DFT (density functional theory) calculations. To conduct the study, we drew upon both previous experimental and theoretical research, and we selected the β-MgCl_2_ surface, which has been shown to best explain the experimental observations. We constructed a three-dimensional (3D) model of the crystallographic structure that matched the standard database in the Materials Project. The crystal in the β phase had a trigonal shape and belonged to the P-3M1 space group, with dimensions a = b = 3.641 Å and c = 5.927 Å. We created layered models, which were three-dimensional (3D) objects repeating in two dimensions, arranged parallel to the specific crystallographic plane defined by its indices (h k l).

The adsorption of inhibitors is a key factor and has a significant impact on the productivity of the Ziegler–Natta catalyst, affecting the selectivity of the catalytic process and the form of the polypropylene (PP) that is produced. In this context, various studies were conducted to understand how these two inhibitors affect the Ziegler–Natta catalyst, evaluating how the interaction of inhibitors with the active titanium sites influences catalytic function.

We investigated the adsorption energy (E_ad_) of these substances using Equation (2). 

In this equation, $E_{cat/Inhibitor}$ represents the total energy of the system, which includes a molecule of methanol or H_2_S bound to the active center of the catalyst (as seen in Figure 11). $E_{cat\;}$ refers to the energy of the catalyst without any molecule of methanol or H_2_S, and the energy of an individual molecule of methanol is $E_{Inhibitor}$. *n* is simply the quantity of methanol or H_2_S molecules that have been adsorbed on the surface.

The results revealed that the *E_ad_* (adsorption energy) for methanol was −39.92 kcal/mol, while for H_2_S, it was −14.82 kcal/mol. These values are remarkable on their own, but what makes our findings even more intriguing is the comparison with previous results by Bahri, who reported an *E_ad_* of −30.6 kcal/mol for methanol in a similar study.

The significant difference in *E_ad_* values between our study and Bahri’s suggests that methanol may have an even greater capacity to adsorb onto the active titanium center than previously considered [37]. This finding could have profound implications in catalysis and catalyst design. The choice of effective inhibitors is crucial for controlling chemical reactions and improving the efficiency of catalytic processes. Our results support the idea that methanol could be a promising option as an inhibitor in specific applications involving the active titanium center, potentially opening new avenues for catalyst optimization in a variety of chemistry and industry fields. These findings are highly relevant and make a significant contribution to current knowledge in the field of catalysis, underscoring the importance of our work in the context of high-impact scientific research.

After investigating how the two inhibitors adhere to the active center of the catalyst, we decided to examine how these molecules adhere to the active center when there is an active TiCl_2_Et compound present. To do this, we used the same group of atoms that we mentioned in the previous section (an active TiIII compound on the (110) surface). This is because it has been suggested that active titanium atoms may be organized on that plane or have a similar environment to what is assumed after they coordinate on that specific plane. A diagram of the model showing how the inhibiting molecules bond to the set containing the active titanium can be seen in Figure 12.

Previous research has established that the bond between inhibitory substances and the titanium center is exceptionally strong compared to other interactions. In a previous study conducted by Bahri, using TiCl_2_Me as the active center of the Ziegler–Natta catalyst, energy values of −27.2, −15.1, −8.4, −13.1, and −30.6 kcal/mol were obtained for interactions with Ti-H_2_O, H_2_S, CO_2_, O_2_, and CH_3_OH, respectively. It is important to note that the interaction with the active titanium proved to be the most favorable among all evaluated [37]. In our study, we focused our attention on interactions with Ti-CH_3_OH and H_2_S using TiCl_2_Et as the active center, and we found energy values of −62.7 and −30.7 kcal/mol, respectively. These results further reinforce the remarkable strength of these interactions.

Our study reveals significantly improved results compared to previous research like Bahri’s. While Bahri obtained energy values of −15.1 and −30.6 kcal/mol for interactions with Ti-H_2_S and CH_3_OH, respectively, our findings show substantially lower values, recording −62.7 and −30.7 kcal/mol for interactions with Ti-CH_3_OH and H_2_S. These results indicate greater stability in the interactions studied in our approach, suggesting a significant advancement in the understanding of these molecular-level reactions.

In Appendix A, we present the cartesian coordinates of our calculations in B3LYP-D3 for Ziegler–Natta catalyst, Methanol, MgCl_2_-TiCl_4_-Methanol interaction, H_2_S, MgCl_2_-TiCl_4_- interaction H_2_S, MgCl_2_/TiCl_4_-Methanol-CH_3_ interaction and MgCl_2_/TiCl_4_-H_2_S-CH_3_ respectively.

### 3.6. Reaction Mechanisms about Methanol and H_2_S as Inhibitors of the Z-N Catalyst

In Figure 13a, the process of polypropylene formation in the absence of inhibitors is explained. In the first step, the monomer is introduced into the π-complex formed between titanium and ethyl. This insertion process occurs through an intermediate state that has a ring structure and leads to the creation of a propyl cation, via an α-agostic interaction. This interaction involves weak bonding between an atom in the monomer molecule and the titanium and ethyl atoms in the complex, partially sharing their electrons. This agostic interaction contributes to the stability of the structure and is a fundamental part of the process that allows the monomer to join the π-complex and form the propyl cation. During this insertion process, the chemical bond between titanium and the ethyl group is broken, allowing titanium to form a new bond with one of the carbon atoms of propylene, while the other carbon atom of propylene establishes a new bond with the ethyl group [27,38].

In a polymerization process, the titanium catalyst (Ti^3+^) in its active state, represented as Ti-CH_2_CH_3_, is essential for the activation of the monomer and its subsequent incorporation into the polymer chain. However, when an inhibitor such as methanol (CH_3_OH) or hydrogen sulfide (H_2_S) is introduced, interactions occur that significantly affect the course of the reaction (see Figure 13b,c). In the case of methanol, the oxygen in methanol is highly chemically attractive and capable of donating a pair of electrons to the titanium atom in the catalyst. This results in the formation of a coordinative bond between the oxygen of methanol and titanium, leaving methanol strongly bound to the active center of the catalyst, as observed in the adsorption energies. This interaction effectively inhibits the arrival of the monomer at the active site, as the oxygen of methanol occupies the space that would normally be available for the polymerization reaction. In the case of H_2_S, sulfur can also coordinate with the titanium atom, forming a coordinative bond similar to that described above. This blocks the active site of the catalyst and prevents the monomer from approaching and binding to the titanium center, halting the polymerization process.

### 3.7. Electronic Properties of the Ziegler–Natta Catalyst

Since titanium in the Ti^4+^ state has no electrons in its d orbitals, in the molecular orbital diagram in Figure 14b, we mainly observe the transfer of electrons from the π orbitals of chlorine to the vacant d orbitals of titanium. This transfer is due to the effect of the crystalline field, which divides the d orbitals into dt2g and deg levels. In a simplified model of a Ti^4+^ surrounded by six chlorine ligands in an octahedral geometry, dt2g orbitals are shown to influence the molecular orbital (MO) diagram significantly. These orbitals mix with the MOs centered on the metal, giving rise to linear combination orbitals adapted to the system’s symmetry. The lowest-energy electronic transitions correspond to the transfer of charge from the π orbitals of chlorine to the dt2g orbitals of titanium (transition A) and the deg orbitals (transition B). However, the experimental spectra do not simply consist of two bands separated by crystal field separation. The d orbitals of Ti^4+^ have a similar symmetry to the π orbitals of the ligands. The π orbitals have lower energy than the d orbitals of the metal, leading to the formation of ligand-centered MO bonds and the occupancy of the bonding t2g MO. In contrast, the antibonding t2g* π orbitals remain vacated.

As a consequence of the interaction between the chlorine ligands and the titanium d orbitals, the metal’s dt2g orbitals experience an increase in energy, leading to a reduction in the crystal field separation (ΔCF) compared to a σ bond. In this framework, the lowest-energy electronic transitions involve the transfer of electrons from the chlorine’s π orbitals to the titanium’s dt2g orbitals (transition A) and the degenerate orbitals (transition B). The computational spectrum derived from the Ziegler–Natta (ZN) catalyst’s structure used in this study (Figure 14a) is consistent with the spectrum of the Ziegler–Natta catalyst modified by the electron donor (ZNC(DBP)) conducted by Piovano et al. [27]. This consistency reveals the presence of the characteristic peak corresponding to the first Cl(π) → Ti(dt2g) transition with a minimum absorption wavelength (band A’, 378 nm). However, it is noteworthy that, in the presence of both inhibitors, a shift in the absorption wavelength is observed, shifting from 377 nm to 399 nm. This phenomenon can be attributed to the interaction of these impurities with the catalyst, thereby influencing its chemical environment. Consequently, a modification in the energy associated with the Cl(π) → Ti(dt2g) transition occurs, resulting in a variation in the absorption wavelength recorded in the UV–vis spectrum. This phenomenon is of significance in industrial applications where precise control of catalytic activity is sought after, necessitating a detailed characterization of the molecular interactions involved.

## 4. Conclusions

Based on the results presented and the detailed analysis of the properties of the obtained polymer, it can be concluded that the presence of hydrogen sulfide (H_2_S) and methanol (CH_3_OH) as inhibitors has a significant impact on the polymerization and the properties of the resulting polymer. A direct correlation was observed between the concentration of H_2_S and methanol and the parameters evaluated, such as the melt index and the average molecular weight of the polymer. As the concentration of these inhibitors increases, there is a decrease in catalyst productivity and an increase in the melt index of the polymer. These effects are due to the interference of the inhibitors in the catalytic activity of the Ziegler–Natta catalyst, which results in an alteration in the length of the polymer chains and a greater fluidity of the polymer. In particular, hydrogen sulfide shows a more pronounced influence compared to methanol. At deficient concentrations, H_2_S causes a drastic decrease in catalyst productivity and a significant increase in the flow rate of the polymer. On the other hand, methanol also negatively affects the molecular weight of the polymer, although its effect is less noticeable than H_2_S.

The theoretical results obtained through calculations based on density functional theory (DFT) provided a detailed understanding of the properties and reactivity of methanol and H_2_S at the molecular level. These calculations allowed us to analyze the HOMO and LUMO orbitals and determine global and local reactivity properties, such as electronegativity, chemical potential, global softness, nucleophilicity, electrophilicity, and hardness indices. Visualization of the distribution of electric charges of the molecules through the molecular electrostatic potential (MEP) map provided additional information about the interactions and the charge distribution. These theoretical results generally improve our understanding of the chemical characteristics and possible reactions of methanol and H_2_S with the Z-N catalyst. All these findings highlight the importance of carefully controlling the presence and concentration of inhibitors during the polymerization process. Close monitoring is required to maintain low H_2_S and methanol concentrations, ensuring optimal properties of the final polymer. In addition, the need to further investigate the underlying mechanisms of inhibitor–catalyst interaction is highlighted to optimize process conditions and minimize adverse effects on polymer production.

## Figures and Tables

**Figure 1 polymers-15-04061-f001:**
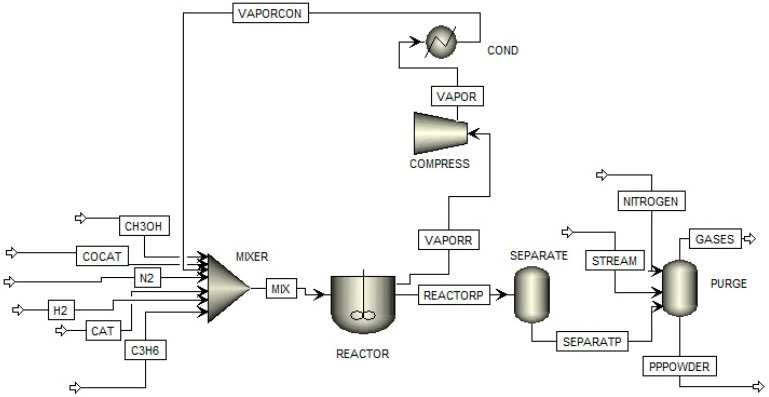
Diagram of the polypropylene production process.

**Figure 2 polymers-15-04061-f002:**
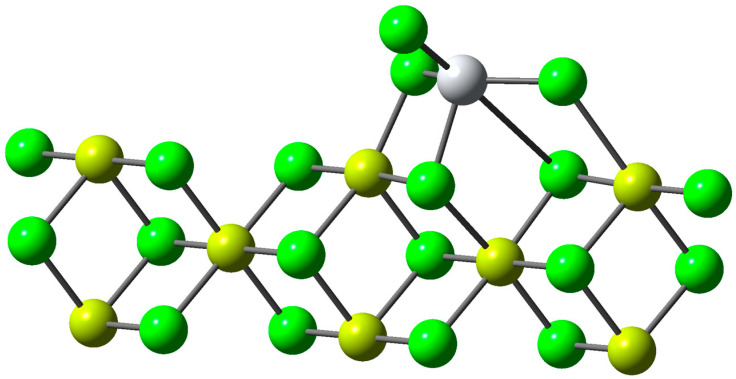
Representative patterns of TiClx species in MgCl_2_ clusters.

**Figure 3 polymers-15-04061-f003:**
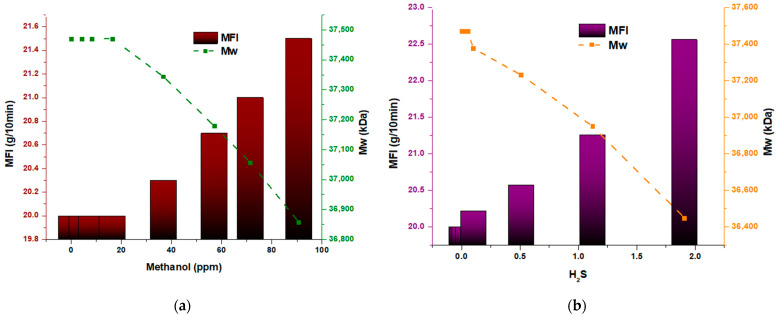
(**a**) Melt flow index and Mw of methanol; (**b**) melt flow index and Mw of H_2_S.

**Figure 4 polymers-15-04061-f004:**
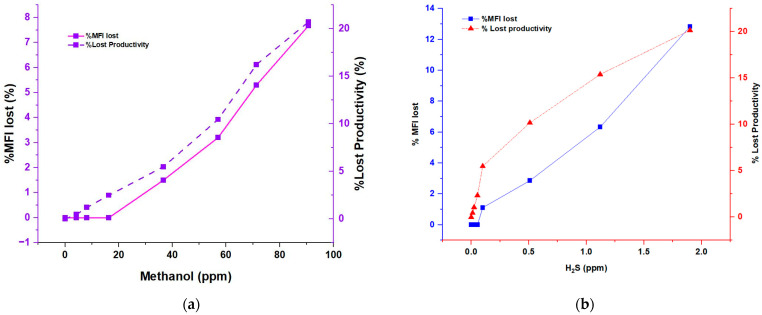
(**a**) Graph of the percentage of MFI lost and loss of catalyst productivity in the presence of methanol; (**b**) graph of the percentage of MFI lost and loss of catalyst productivity in the presence of H_2_S.

**Figure 5 polymers-15-04061-f005:**
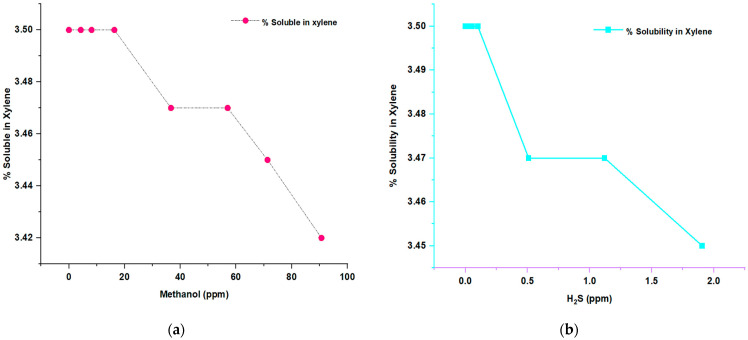
(**a**) Graph of the solubility in xylene and the selectivity control agent in the presence of methanol; (**b**) graph of the solubility in xylene and the selectivity control agent in the presence of H_2_S.

**Figure 6 polymers-15-04061-f006:**
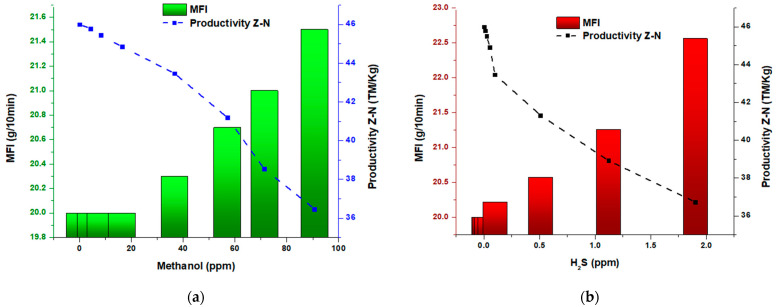
(**a**) Graph of MFI vs. Productivity of the Ziegler–Natta catalyst in methanol bulk; (**b**) graph of MFI vs. Productivity of the Ziegler–Natta catalyst in H_2_S presence.

**Figure 7 polymers-15-04061-f007:**
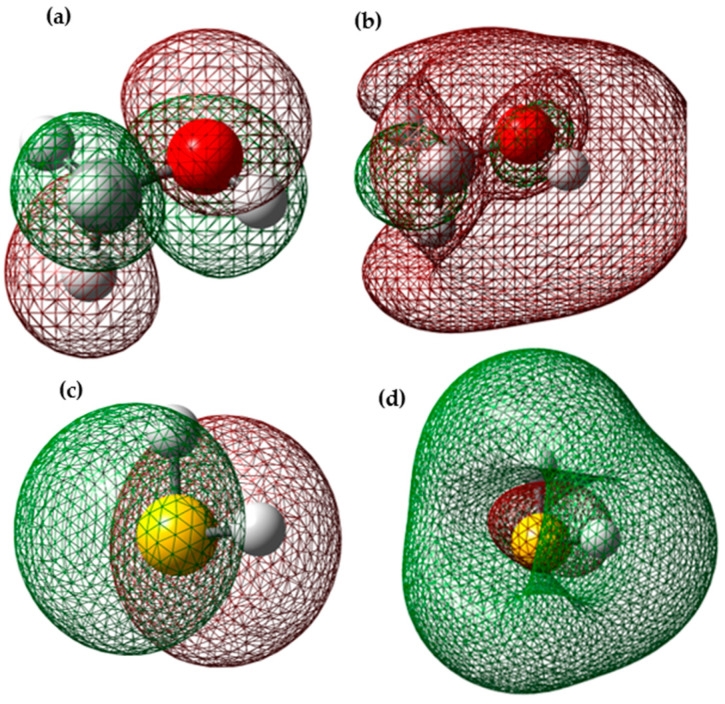
(**a**) Highest occupied molecular orbital (HOMO) of methanol; (**b**) lowest unoccupied molecular orbital (LUMO) of methanol; (**c**) highest occupied molecular orbital (HOMO) of H_2_S; (**d**) lowest unoccupied molecular orbital (LUMO) of H_2_S.

**Figure 8 polymers-15-04061-f008:**
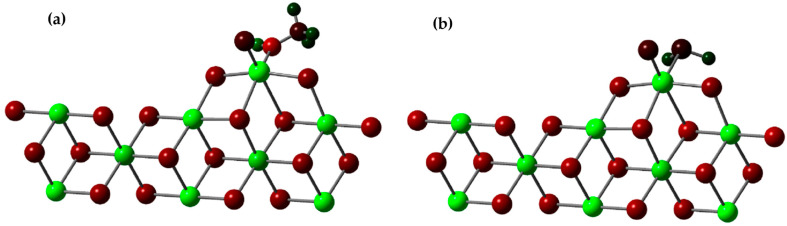
(**a**) Charge distribution of adsorbed methanol on the Ziegler–Natta catalyst; (**b**) charge distribution of adsorbed H_2_S on the Ziegler–Natta catalyst.

**Figure 9 polymers-15-04061-f009:**
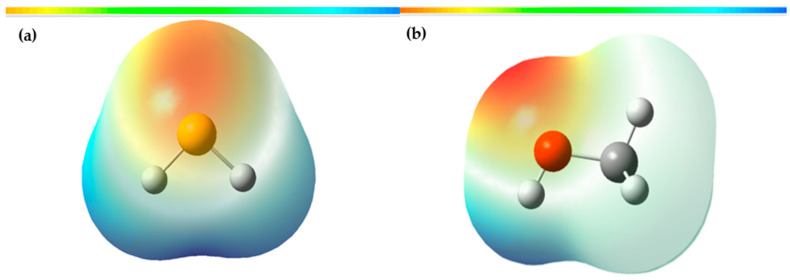
(**a**) Total density mapped to the electrostatic surface of H_2_S; (**b**) total density mapped to the electrostatic surface of methanol.

**Figure 10 polymers-15-04061-f010:**
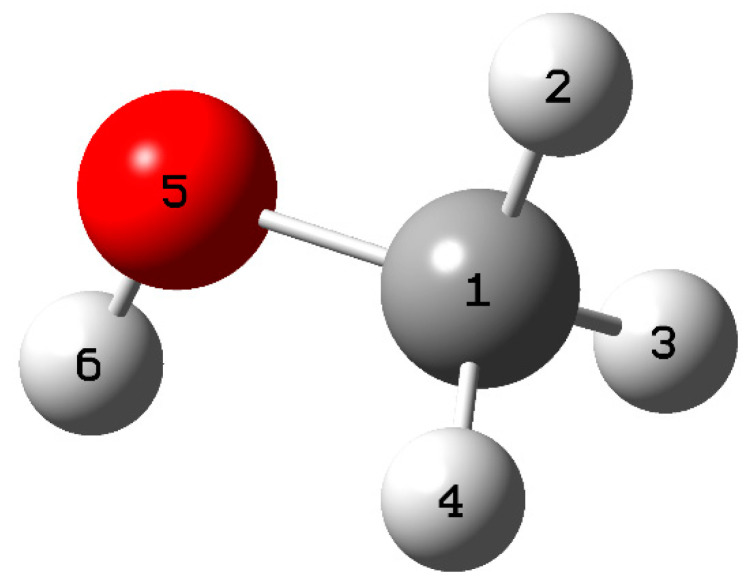
Spatial conformation of methanol.

**Figure 11 polymers-15-04061-f011:**
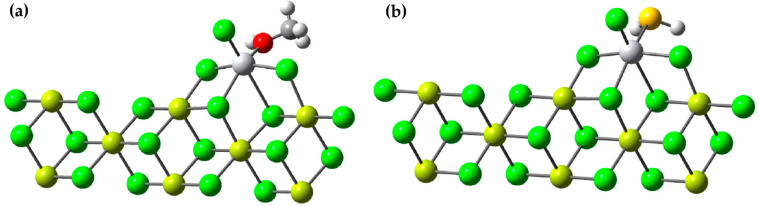
(**a**) Adsorption of a molecule of methanol onto the active center of the Ziegler–Natta catalyst; (**b**) adsorption of a molecule of H_2_S onto the active center of the Ziegler–Natta catalyst. The chlorine atoms are identified by the color green, the magnesium in yellow, the oxygen atom in red, the sulfur atom in ocher yellow, the titanium atom in gray and the carbon atoms in light gray.

**Figure 12 polymers-15-04061-f012:**
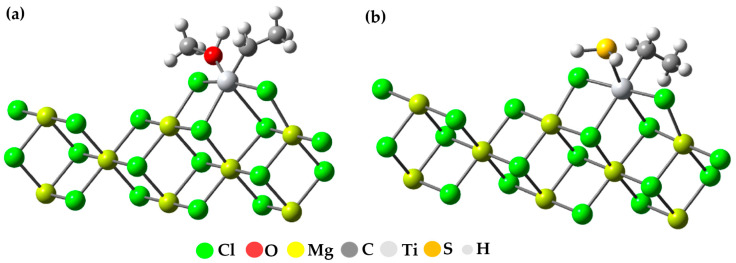
(**a**) Structure of the methanol binding to titanium in the Mg_8_Cl_16_-Ti CH_2_CH_3_Cl_2_ model; (**b**) structure of the H_2_S binding to titanium in the Mg_8_Cl_16_-Ti CH_2_CH_3_Cl_2_ model.

**Figure 13 polymers-15-04061-f013:**
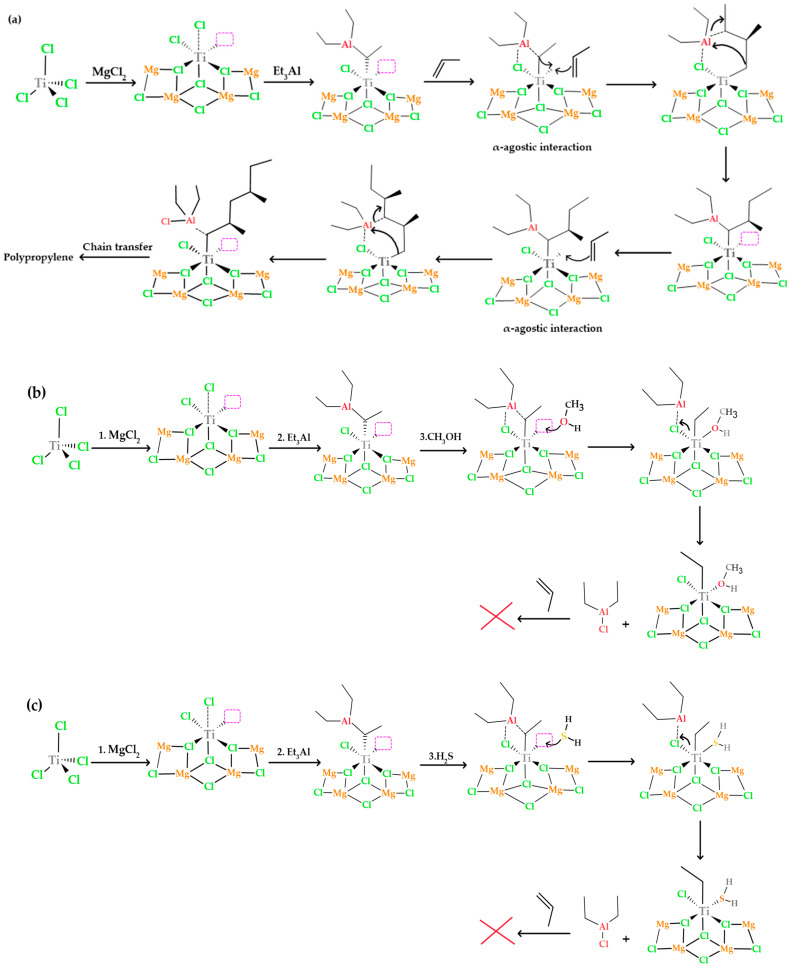
(**a**) Polymerization mechanism of propylene in the presence of triethylaluminum and Ziegler–Natta catalyst; (**b**) mechanism of methanol as a Ziegler–Natta catalyst inhibitor; (**c**) reaction mechanism of H_2_S in TiCl_4_/MgCl_2_.

**Figure 14 polymers-15-04061-f014:**
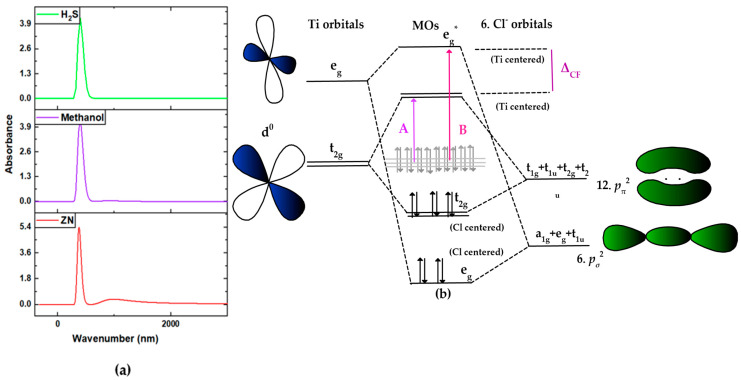
(**a**) UV–vis spectrum of the Ziegler–Natta catalyst; (**b**) simplified molecular orbital diagram for a Ti^4+^ in an octahedral field surrounded by six Cl with sigma–ligand bonds.

**Table 1 polymers-15-04061-t001:** Quantities of methanol and H_2_S collected during the polymerization process and measurements of MFI, Mw, and % soluble in xylene.

**Methanol (ppm)**	0	0	0	4.3	4.2	4.3	8.1	8.2	8.1	16	16	17	37	36	37	57	58	56	71	72	71	91	90	91
**TM of PP produced**	46	46	46	45.86	45.74	45.76	45.53	45.37	45.43	44.85	44.84	44.87	43.47	43.46	43.49	41.23	41.15	41.19	38.57	38.45	38.57	36.56	36.49	36.36
**Productivity Ziegler–Natta (TM/Kg)**	46	46	46	45.86	45.74	45.76	45.53	45.37	45.43	44.85	44.84	44.87	43.47	43.46	43.49	41.23	41.15	41.19	38.57	38.45	38.57	36.56	36.49	36.36
**% Productivity loss**	0	0	0	0.3	0.57	0.52	1.02	1.37	1.24	2.5	2.52	2.46	5.5	5.52	5.46	10.37	10.54	10.46	16.15	16.41	16.15	20.52	20.67	20.96
**MFI**	20	20	20	20	20	20	20	20	20	20	20	20	20.3	20.3	20.3	20.7	20.7	20.7	21	21	21	21.5	21.5	21.5
**% MFI loss**	0	0	0	0	0	0	0	0	0	0	0	0	1.5	1.5	1.5	3	3.3	3.3	5.3	5.3	5.3	7.6	7.7	7.7
**% Soluble in xylene (%Xs)**	3.5	3.5	3.5	3.5	3.5	3.5	3.5	3.5	3.5	3.5	3.5	3.5	3.47	3.48	3.48	3.47	3.47	3.47	3.45	3.45	3.45	3.43	3.42	3.43
**H_2_S (ppm)**	0.013	0.012	0.013	0.025	0.025	0.026	0.054	0.05	0.06	0.1	0.11	0.104	0.51	0.505	0.512	1.1	1.12	1.14	1.91	1.905	1.906	0.013	0.012	0.013
**TM of PP produced**	45.77	45.79	45.82	45.48	45.51	45.55	44.92	44.96	44.89	43.5	43.48	43.45	41.35	41.33	41.29	38.95	38.89	38.92	36.78	36.72	36.69	45.77	45.79	45.82
**Productivity Ziegler–Natta (TM/Kg)**	45.77	45.79	45.82	45.48	45.51	45.55	44.92	44.96	44.89	43.5	43.48	43.45	41.35	41.33	41.29	38.95	38.89	38.92	36.78	36.72	36.69	45.77	45.79	45.82
**% Productivity loss**	0.5	0.46	0.39	1.13	1.07	0.98	2.35	2.26	2.41	5.43	5.48	5.54	10.11	10.15	10.24	15.33	15.46	15.39	20.04	20.17	20.24	0.5	0.46	0.39
**MFI**	20	20	20	20	20	20	20	20	20	20.24	20.23	20.2	20.57	20.55	20.6	21.3	21.3	21.2	22.6	22.5	22.6	20	20	20
**% MFI loss**	0	0	0	0	0	0	0	0	0	1.2	1.15	1	2.85	2.75	3	6.5	6.5	6	13	12.5	13	0	0	0
**% Soluble in xylene (%Xs)**	3.5	3.5	3.5	3.5	3.5	3.5	3.5	3.5	3.5	3.46	3.47	3.47	3.47	3.47	3.47	3.45	3.45	3.45	3.44	3.44	3.44	3.5	3.5	3.5

**Table 2 polymers-15-04061-t002:** Global chemical reactivity descriptors for H_2_S and methanol calculated using B3LYP/6–311G(d,p).

Parameters	H_2_S	Methanol
η = ½ (E_LUMO_ − E_HOMO_) (eV)	3.9176	4.4949
χ = −1/2 (E_LUMO_ + E_HOMO_) (eV)	3.3221	2.6698
S = 1/η (eV)	0.2552	0.2225
µ = 1/2(E_LUMO_ + E_HOMO_) (eV)	−3.3221	−2.6698
ω = µ^2^/2η (eV)	1.4086	0.7929
N = E_HOMO(DFM)_ − E_HOMO(TCE)_ (eV)	1.9069	1.9961

**Table 3 polymers-15-04061-t003:** Quantum chemical descriptors for H_2_S and methanol calculated using B3LYP/6–311G(d,p).

Parameters	H_2_S	Methanol
E_HOMO_ (eV)	−7.254	−7.1648
∆E (eV)	7.8352	8.9899
E_LUMO_ (eV)	0.6098	1.8251

**Table 4 polymers-15-04061-t004:** Fukui functions for H_2_S.

Number	$f^{+}$	$f^{-}$	$f^{0}$	∆f
1	0.7080	1.0000	0.8540	−0.2920
2	0.1436	0.0000	0.0718	0.1436
3	0.1484	0.0000	0.0742	0.1484

**Table 5 polymers-15-04061-t005:** Fukui functions for methanol.

Number	$f^{+}$	$f^{-}$	$f^{0}$	∆f
1	0.2479	0.0313	0.1396	0.2166
2	0.0235	0.0000	0.0118	0.0235
3	0.0537	0.0792	0.0665	−0.0255
4	0.0537	0.0792	0.0665	−0.0255
5	0.5302	0.8103	0.6702	−0.2800
6	0.0909	0.0000	0.0454	0.0909

## Data Availability

Not applicable.

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
