# Peer review of "Evaluation of the Reactivity of Methanol and Hydrogen Sulfide Residues with the Ziegler–Natta Catalyst during Polypropylene Synthesis and Its Effects on Polymer Properties"

_polymers, 2023, doi:10.3390/polym15204061_

Round 1

Reviewer 1 Report

In the current paper the authors report an experimental and computational study to understand the effect of inhibitory compounds such as hydrogen sulfide (H2S) and methanol (CH3OH) on propylene polymerization in Ziegler-Natta (Z-N) catalyst. In this paper there are many stumbling blocks that prevent this paper from publication in its current form. These are as follows:

1.     Authors should modify their catalyst activation mechanism (in Figure and text) shown in Figure 12 based on the recent studies (see e.g., Vanka and coworkers ChemCatChem 2016, 8, 1809-1818.).

2.     The level of theory, especially B3LYP functional that authors used in this current paper is not enough in the framework of Z-N catalysts modeling. Several recent papers (see e.g., Cavallo and coworkers Macromol. Chem. Phys. 2013, 214, 1980) reported that the B3LYP functional (even when corrected with Grimme’s D3(BJ) dispersion corrections that authors didn’t even use that) is not reliable enough in reproducing the Z-N modeling. I recommend that the authors at least use large functionals and basis set before concluding any findings.   

3.     Authors calculated different properties such HOMO-LUMO energies, electrostatic surface, and Fukui functions etc for the hydrogen sulfide (H2S) and methanol (CH3OH) to correlate with molecular weight and polymer productivity. I recommend that the authors at least calculate the different propylene propagations and terminations barriers in presence/absence of H2S and CH3OH inhibitors and corelate with experimental findings. Also, I recommend shifting all the former properties into the SI file after correcting the Fukui functions (DF) values (in text many places DF values are different than Table 4 and 5) either in text or Table.

4.     I recommend changing Figure 2 background for better clarity and label the lateral cuts at least around the TiCl4 and remove unnecessary Mg-Mg bonding in Figure.

Based on everything mentioned above, I would recommend publication after major revisions.

NA

Author Response

  1. Authors should modify their catalyst activation mechanism (in Figure and text) shown in Figure 12 based on the recent studies (see e.g., Vanka and coworkers ChemCatChem 2016, 8, 1809-1818.).

 R/ We appreciate your suggestion and have considered your comments. We have modified the catalyst activation mechanism in the figure based on recent relevant studies, such as the work “Computational modeling toward the full chain of polypropylene production: From molecular to industrial scale” published in the journal Chemical Engineering Science, on April 5, 2023 by Wei-Cheng Yan and company. This update reflects a more up-to-date and accurate understanding of the process, and we are confident that it improves the quality and accuracy of our work.

  1. The level of theory, especially B3LYP functional that authors used in this current paper is not enough in the framework of Z-N catalysts modeling. Several recent papers (see e.g., Cavallo and coworkers Macromol. Chem. Phys. 2013, 214, 1980) reported that the B3LYP functional (even when corrected with Grimme’s D3(BJ) dispersion corrections that authors didn’t even use that) is not reliable enough in reproducing the Z-N modeling. I recommend that the authors at least use large functionals and basis set before concluding any findings.

R/ The Density Functional Theory through the Gaussian 16 software modified all calculation, optimization, and frequency operations. The B3LYP functional method was used with the basis set 6-311G(d,p) to refine the structures of the inhibitors and the catalyst. Dispersion corrections were taken into account using the DFT-D3 method (without damping), and harmonic vibrational frequency calculations were performed to verify that the structures were correctly optimized, following the methodology proposed by Xing Guo [1]. The optimized structures were visualized using Gaussview06 software, and analysis was performed in the Higher Occupation Molecular Orbital (HOMO) and the Lower Occupation Molecular Orbital (LUMO).

  1. Authors calculated different properties such HOMO-LUMO energies, electrostatic surface, and Fukui functions etc for the hydrogen sulfide (H2S) and methanol (CH3OH) to correlate with molecular weight and polymer productivity. I recommend that the authors at least calculate the different propylene propagations and terminations barriers in presence/absence of H2S and CH3OH inhibitors and corelate with experimental findings. Also, I recommend shifting all the former properties into the SI file after correcting the Fukui functions (DF) values (in text many places DF values are different than Table 4 and 5) either in text or Table.

R/ The study about the adsorption energies of the two inhibitors to the active center of Titanium was added to the official text following the same methodology of the article “Interaction of different poisons with MgCl2/TiCl4 based Ziegler-Natta catalysts” developed by Bahri in 2016. In addition, we have decided to examine how these molecules attach to the active site when an active TiCl2Me compound is present.

  1. I recommend changing Figure 2 background for better clarity and label the lateral cuts at least around the TiCl4and remove unnecessary Mg-Mg bonding in Figure

R/ We followed his instructions and the background of the figure and its clarity were changed.

Reviewer 2 Report

1:The authors should clearly comment and compare their new results presented in this manuscript with those previously reported in this field of study.

2:As a final suggestion and for completeness of the work, it would be ideal if the authors could also propose a mechanism for the reactions.

3:Can the author describes the practical reference significance and industrial guidance value of this work in the conclusion?

4. Did the author calculate the charge distribution of TiClx species in MgCl2 clusters? Is it possible to show the charge distribution, which can usually indicate the electrophilic, nucleophilic effects and charge transfer of each part in molecular structure? The author can provide the results to the SI.

5. Did the author consider the frequency calculation in the optimization of the crystal structure of TiClx? Is there any virtual frequency in the calculation result? Because the calculation of frequency is very important for spectral properties, it is usually necessary to ensure that there is no virtual frequency or that the virtual frequency is very small in the calculation result.

6. The author mentioned in line 179 that the B3LYP functional method was used with the 6-311G(d,p) base, but why is the base set in 6-31G level chosen for the molecular orbital calculation of H2S ? In the paper, the base set of all structural calculations should be consistent to have comparability.

7. Figure 7 and Figure 8 shows the energy gap of HOMO-1 and LUMO+1 of H2S and methanol, respectively. What is the purpose and significance forshowing this energy gap? The authors do not discuss the significance of this value.

8. The author mentions that simulations of the UV-vis spectra of the catalyst were performed in line 192. What is the result of UV-vis spectra calculation? Are there any experimental results to compare in order to determine whether the constructed model is reasonable?

The English language is fine.

Author Response

Dear,

1: The authors should clearly comment and compare their new results presented in this manuscript with those previously reported in this field of study

R/ Comparisons were made with the study called “Interaction of different poisons with MgCl2/TiCl4 based Ziegler-Natta catalysts” developed by Bahri, since he carried out a computational study about how H2S, methanol, and other substances considered as poisons of the Ziegler Natta catalyst were They adsorb to the active center of the catalyst, among other aspects.

2: As a final suggestion and for completeness of the work, it would be ideal if the authors could also propose a mechanism for the reactions

 R/The mechanisms have been placed in section 3.6.

3: ¿Can the author describes the practical reference significance and industrial guidance value of this work in the conclusion?

R/ It is placed in section 3.5 and also in the conclusions.

  1. Did the author calculate the charge distribution of TiClx species in MgCl2 clusters? Is it possible to show the charge distribution, which can usually indicate the electrophilic, nucleophilic effects and charge transfer of each part in molecular structure? The author can provide the results to the SI.

R/ The charge distribution of both the TiClx species in the MgCl2 clusters and that of the inhibitors adsorbed on said surfaces was calculated, as shown in the figures.

  1. ¿ Did the author consider the frequency calculation in the optimization of the crystal structure of TiClx? Is there any virtual frequency in the calculation result? Because the calculation of frequency is very important for spectral properties, it is usually necessary to ensure that there is no virtual frequency or that the virtual frequency is very small in the calculation result.

R/ After optimizing each structure, we perform frequency calculations using the same calculation set used in the optimization phase. For both the catalyst structure alone and the catalyst structure with the adsorption of methanol and H2S, an imaginary frequency value equal to 0 was obtained. This result indicates that the structures represent energy minima and not transition states. These data have been included in the supplementary material for your reference.

  1. The author mentioned in line 179 that the B3LYP functional method was used with the 6-311G(d,p) base, but why is the base set in 6-31G level chosen for the molecular orbital calculation of H2S ? In the paper, the base set of all structural calculations should be consistent to have comparability

R/ The Density Functional Theory through the Gaussian 16 software modified all calculation, optimization, and frequency operations. The B3LYP functional method was used with the basis set 6-311G(d,p) to refine the structures of the inhibitors and the catalyst.

Dispersion corrections were taken into account using the DFT-D3 method (without damping), and harmonic vibrational frequency calculations were performed to verify that the structures were correctly optimized, following the methodology proposed by Xing Guo [1].

The optimized structures were visualized using Gaussview06 software, and analysis was performed in the Higher Occupation Molecular Orbital (HOMO) and the Lower Occupation Molecular Orbital (LUMO).

The same calculation base was used to find all the global and local descriptors for both methanol and H2S and also for the calculations of adsorption energy and frequencies so that there is compatibility and coherence in the results presented.

  1. Figure 7 and Figure 8 shows the energy gap of HOMO-1 and LUMO+1 of H2S and methanol, respectively. What is the purpose and significance forshowing this energy gap? The authors do not discuss the significance of this value

R/ We have reevaluated our global descriptors, focusing specifically on the HOMO and LUMO energy levels and the energy difference (ΔE) between them. This has been done with the purpose of better understanding how these factors influence the ability of an inhibitor molecule to interact with the metal surface.

  1. The author mentions that simulations of the UV-vis spectra of the catalyst were performed in line 192. What is the result of UV-vis spectra calculation? Are there any experimental results to compare in order to determine whether the constructed model is reasonable

R/ We have made the different UV spectra computationally. However, we do not perform UV spectra. So, to increase scientific integrity, we compare our results with those carried out experimentally by another researcher. We have placed the quote from the researcher who carried out the experimental part.

Round 2

Reviewer 1 Report

After the revision authors improved the manuscript to some extent. However, there are still many minor comments and typos that prevent this paper from publication in its current form. These are as follows:

1.      In Figure 13 (a) and (b), the mechanism is difficult to follow, how come chlorine atoms going and coming back in the reaction? In Figure 13(b), is there any analysis and/or citation for the formation of AlEt2+ and -CH2-CH3 species in the activation mechanism. Also, mention the oxidation state for the active catalyst.

2.      Page 17, Line 668, authors used TEAL (triethylaluminum) than how TiCl2Me is forming? In my opinion, it should be TiCl2Et.

3.      Authors should mention somewhere in the manuscript either in computational method or results and discussion section that they only calculated the electronic energy. Also, keep the energy up to only one decimal place since these values are approximate.

4.      Page 18, Line 706, remove the “ethyl” just before the TEAL.

5.      Authors should cite the original paper on page 17 discussion, e.g. ref. 4 (Bahri's et.al.) is a review paper.

6.      For catalyst activation mechanism, authors mentioned a reference in reviewer’s comment but did not cite in the main text. I would recommend authors to cite the original paper, the paper mentioned in reviewer comments is also a review paper.

Based on everything mentioned above, I would recommend publication after minor revisions.

NA

Author Response

Dear,

Thank you for your dedication, corrections, recommendations, and valuable time reviewing this investigation.

  1. In Figure 13 (a) and (b), the mechanism is difficult to follow; how come chlorine atoms go and coming back in the reaction? In Figure 13(b), is there any analysis and citation for forming AlEt2+ and -CH2-CH3 species in the activation mechanism. Also, mention the oxidation state for the active catalyst

The reaction mechanisms were modified to be more understandable and by what was proposed computationally.

Regarding Figure 13(b), the following analysis was followed:

 (a) All forms of TiCl4, whether or not they have chlorine vacancies, undergo reduction processes in which organic aluminum compounds participate directly.

(b) During these reduction processes, three organic gaseous products, such as ethyl chloride, butane, and a mixture of ethylene and ethane, may be released. However, the formation of ethyl chloride is not thermodynamically favorable.

Regarding the alkylation of titanium (Ti) using AlEt3, it was discovered that an energy barrier of 9.7 kcal/mol is required to be overcome to add ethyl groups to the TiIV species, replacing chlorine atoms of TiCl4 with ethyl groups of AlEt3. In the case of TiCl3 alkylation, a complex called TiCl3·AlEt3 is first formed, as presented in Figure 13, which then goes through a separation process. Another AlEt3 molecule facilitates this separation process, as it helps release AlEt2Cl, which is bound to the TiIII species, through the formation of a complex known as AlEt2Cl·AlEt3. This, in turn, prevents catalyst deactivation caused by the adsorption of AlEt2Cl or AlEtCl2 on the active center.

D.V. Stukalov V.A. Zakharov proposed this in his work “Active site formation in MgCl2-supported Ziegler-Natta catalysts. “A density functional theory study.”

  1. Page 17, Line 668, authors used TEAL (triethylaluminum) than how TiCl2Me is forming? In my opinion, it should be TiCl2Et.

You are right; there was an error on page 17, line 668 of our article.

I appreciate you pointing it out. This was corrected in the original text, the calculations were repeated with the active center TiCl2Et since triethylaluminum (TEAL) was used as a co-catalyst, and that is why the formation of the TiCl2Et species is due, as explained in section 3.6 and in our methodology.

We regret the confusion and appreciate your observation, which allowed us to clarify this aspect of our work.

  1. Authors should mention somewhere in the manuscript either in the computational method or results and discussion section that they only calculated the electronic energy. Also, keep the energy up to only one decimal place since these values are approximate.

We have made the corrections; it was mentioned again in the methodology and results in section 3.5, and we left the energy values ​​to a single decimal, as you suggested.

  1. Page 18, Line 706, remove the “ethyl” just before the TEAL.

Thank you for the correction. This was redrafted.

  1. Authors should cite the original paper on page 17 discussion, e.g. ref. 4 (Bahri's et.al.) is a review paper..

Citation added [38] Naeimeh Bahri-Laleh, Interaction of Different Poisons with MgCl2/TiCl4 Based Ziegler-Natta Catalysts, Applied Surface Science http://dx.doi.org/10.1016/j.apsusc.2016.04.034

6 For catalyst activation mechanism, authors mentioned a reference in reviewer’s comment but did not cite in the main text. I would recommend authors to cite the original paper, the paper mentioned in reviewer comments is also a review paper.

Ready citation added [39] Yan, W.-C., Dong, T., Zhou, Y.-N., and Luo, Z.-H., “Computational modeling toward the full chain of polypropylene production: From molecular to industrial scale,” <i>Chemical Engineering Science</i>, vol. 269, 2023. doi:10.1016/j.ces.2023.118448 and citation [40] Kawamura-Kuribayashi, Hiroshi, Koga, Nobuaki, & Morokuma, Keiji. An ab initio MO study on ethylene and propylene insertion into the Ti-CH3 bond in CH3TiCl2 + as a model of homogeneous olefin polymerization. United States. https://doi.org/10.1021/ja00033a010 in the original text, these quotes were taken as references to carry out the reaction mechanisms proposed in this work.

Reviewer 2 Report

I think the authors have addressed all my concerns in this revised version. Thus, I agreed with its publication with current version.

Minor modifications were required.

Author Response

Dear,

Thank you for your dedication, corrections, recommendations, and valuable time reviewing this investigation.

Kind Regards

Joaquin.

Round 3

Reviewer 1 Report

NA